# Sustainability Assessment of a Low-Income Building: A BIM-LCSA-FAHP-Based Analysis

Marcus V. A. P. M. Filho [1], Bruno B. F. da Costa [2], Mohammad Najjar [3], Karoline V. Figueiredo [3], Marcos Barreto de Mendonça [3] and Assed N. Haddad [3,*]

1   Programa de Engenharia Civil, Universidade Federal Fluminense, Niterói 24210-240, Brazil; marcusplaisant@id.uff.br
2   Instituto Politécnico, Universidade Federal do Rio de Janeiro, Macaé 27930-560, Brazil; bruno.barzellay@macae.ufrj.br
3   Programa de Engenharia Ambiental, Universidade Federal do Rio de Janeiro, Rio de Janeiro 21941-901, Brazil; mnajjar@poli.ufrj.br (M.N.); karolinefigueiredo@poli.ufrj.br (K.V.F.); mbm@poli.ufrj.br (M.B.d.M.)
*   Correspondence: assed@poli.ufrj.br

**Abstract:** The construction industry is one of the most significant consumers of environmental resources worldwide. Faced with the need to produce new buildings, but without further burdening the environment, attempts to improve social, economic, and environmental indicators have turned attention to building construction in recent decades. The objective of this research is to develop a novel framework to assess the most sustainable choice of materials applied to the construction of low-income buildings, according to the three pillars of the Triple Bottom Line (TBL). A BIM-LCSA-FAHP-based model was proposed with the creation of nine different scenarios, where the materials of the structure (precast concrete, cast-in-place concrete, and structural masonry), painting (PVA water-based and acrylic), and roofing (ceramic and fiber cement tiles) varied. The proposed procedure consists of the elaboration of a 3D Building Information Modeling (BIM) model, for which the parameters described above were evaluated according to the Life Cycle Sustainability Assessment (LCSA)-TBL-based criteria, divided into ten sub-criteria, that includes: (1) environmental (acidification, eutrophication, global warming, ozone depletion, smog formation, primary energy, non-renewable energy, and mass total), (2) economic (construction cost) and (3) socio-political issues (community impact). Finally, the Fuzzy Analytical Hierarchy Process (AHP) was used as a multi-criteria decision-making technique that helps in aggregating and classifying the impacts of each scenario in a sustainability index (*SI*). Regarding the best option for low-income construction, the results indicated that precast concrete when combined with acrylic paint and fiber cement tiles (scenario 3) proved to be the most advantageous and achieved first place in the sustainability index (*SI*) developed in this work. This methodology is replicable for different construction typologies and several categories of materials, making it a robust decision-aiding tool for engineers, architects, and decision makers.

**Keywords:** triple bottom line; life cycle sustainability assessment; building information modeling; multicriteria decision making; fuzzy analytic hierarchy process; sustainability index; sustainable building

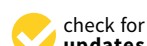



## 1. Introduction

The construction industry is one of the most significant consumers of environmental resources worldwide [1,2], and one of the biggest industries responsible for giving rise to large amounts of waste [3]. The construction sector uses 30–40% of all-natural resources and primary energy over its lifespan, accounting for 30% of global greenhouse gas (GHG) emissions and representing about 6% of the world's Gross Domestic Product (GDP) [4,5]. Despite the sector's relevance in the international economy, the United Nations (UN) estimates that about one billion people globally still live in inadequate buildings or do not have a place to live [6]. Access to adequate housing is an internationally recognized

basic human right that plays an important role in society, providing citizens with dignity and security [7,8]. In Brazil, the housing deficit is approximately 6.2 million dwellings, which mainly affects low-income families [9–11]. Therefore, in an attempt to mitigate this social problem and provide access to housing for the portion of the population traditionally excluded from the conventional real estate market, the Brazilian Federal Government created the program designated "Minha Casa, Minha Vida" (My House, My Life), aimed at simplifying, financing, and encouraging the construction and acquisition of low-cost houses [10,12,13].

Low-income buildings or social housing can be defined as dwellings built on a large scale with government funding aimed at low-income populations [9]. However, due to the growing demand for this type of building, decision makers are often pressured to speed up the construction process and reduce costs [10]. As a result, there is a tendency to use lower-cost materials, which are not always the most beneficial choices for the user and the environment [11]. Thus, it is essential to develop material-selection tools that help decision making in a transparent, reliable, and sustainable way. In this context, faced with the need to produce new buildings, but without further burdening the environment, attempts to improve social, economic, and environmental indicators have turned attention to building construction. These efforts have focused on complying with the pillars of the theory known as Triple Bottom Line (TBL), which considers that development only actually occurs when the best use of natural resources, the guarantee or improvement of the current economic balance, and the occurrence of social gains are observed concomitantly and with parity [14,15]. This theory, formulated more than two decades ago, gained notoriety by providing a framework for evaluating the sustainability of products and services based on criteria other than just the environment. Currently, additional dimensions have been added to the original concept, such as the cultural pillar and the political pillar, but the original tripod still symbolizes the basic elements to be considered in a sustainability assessment. Unfortunately, this strategy is not always used in the construction market. Environmental and social aspects are usually neglected, and relevant decisions are taken by considering mainly the economic aspects, which are more easily measured.

Building sustainability is a plural and complex topic. However, there is a consensus that proper design, including the suitable selection of construction techniques and building materials to be used, can improve constructions performance throughout their life cycle [16]. In this context, there is a great potential to reduce GHG emissions in buildings, also causing positive socioeconomic impacts [17]. The subject has been widely discussed in recent decades [18], and several concepts, tools, and methodologies have been developed in order to achieve the main objectives of sustainable construction, such as Building Information Modeling (BIM) [19], Multicriteria Decision Making (MCDM) [1], Life Cycle Analysis (LCA) [20], and Life Cycle Sustainability Assessment (LCSA) [16]. However, these techniques are generally applied in isolation, which is not consistent with the dynamic and multifaceted context of the construction sector, where several decisions can directly impact the different aspects of a project. Thus, architects, engineers, and decision makers often have difficulty in defining which aspects should be prioritized when planning a project, as well as which items are more relevant in the context of TBL theory [21]. Therefore, considering that in countries where construction is at a high-maturity stage, much of the total work schedule is used in the planning and design phase, this work proposes a novel framework for building-material selection using the TBL's sustainability criteria.

The main objective of this study is to provide a comprehensive and systemic decision-making tool simultaneously based on BIM, LCSA, and MCDM, which uses the best environmental, economic, and social building-material choice when planning a new construction project. The methodology was validated through a case study in a low-income building where nine scenarios with different configurations of building materials were simulated in a BIM software. TBL's sustainability criteria were addressed through the LCSA technique, which assesses the social, economic, and environmental positive and negative impacts and benefits of a service or product along this life cycle [16]. This approach was chosen

because LCSA overcomes the LCA [22]. Finally, the Fuzzy Analytic Hierarchy Process (FAHP), a well-known MCDM technique, was used to define the construction model that best fits the TBL criteria. This study will assist decision makers, stakeholders, and builders in applying the best materials and construction methodologies, especially in developing large housing programs, where the resources invested and their respective impacts are substantially significant.

The remainder of this paper is structured as follows: Section 2 presents the conceptual background of the research, based on a review of the previous studies on LCSA, MCDM methods, and BIM. Section 3 describes the scenarios created in BIM, the fuzzy-AHP approach, and the LCSA overall parameters employed in this study. Section 4 presents the case study to verify the proposed model and the research findings. Finally, Section 5 discusses study implications, summarizes the conclusions, and exposes work limitations and directions for further research.

## 2. Literature Review

The impact caused by the construction industry on the environment, economy, and society is significant [23]. The large number of resources required for the construction and retrofit of buildings gradually brought sustainable construction into focus [24]. In this sense, the selection of suitable and sustainable building materials during the design stage has a direct impact on the entire life cycle of the building [25,26], promoting the reduction of its cost, allowing a better use of natural resources, reducing the generation of waste during construction and operation stages, optimizing energy consumption, and providing a healthier environment for users [27,28]. Therefore, several researchers conclude that the correct selection of materials is one of the simplest and most efficient ways to incorporate sustainability in the construction industry [25,29,30]. Despite the relevance of optimizing the construction materials selection process, currently, in practice, this is still primarily done empirically, based on the engineers' knowledge and experience, rather than numerical approaches [30–32]. However, when associated with the lack of experience of the responsible professional, this heuristic approach prevents the available alternatives from being objectively compared [25,33], and the lowest price ends up prevailing as the most important choice criterion. Therefore, it is essential to develop a systematic decision model that helps team members to choose the most advantageous materials, based on well-defined, sustainable, and unambiguous criteria [23,32].

In order to overcome this problem, several methodological approaches have been employed to deal with the material selection problem [24,27]. Some recent studies have used the the Building Research Establishment Environmental Assessment Method (BREEAM) as a construction material selection tool, while others proposed the use of the Leadership in Energy and Environmental Design (LEED) rating system [34,35]. However, Life Cycle Assessment (LCA) is still the most applied methodology to assess the impacts of a product throughout its entire life cycle [36]. Takano et al. [37] used LCA to research the influence of material selection on the energy balance of a building. Najjar et al. [20] studied the performance of building materials using a BIM-LCA approach. Gardner et al. [38] conducted a material LCA in a building comparing the impacts of each stage of the construction process. Röck et al. [39] used a BIM-LCA-integrated approach to compare building elements contribution to the total embodied environmental impact. Abd Rashid et al., [40] evaluated a cradle-to-grave environmental impact of a residential building in Malaysia through LCA. However, LCA-based models focus on evaluating the environmental performance of a product, while not adequately considering issues related to its costs and society [24,30,41], making it necessary to develop new methods that overcome these flaws, integrating the three pillars of sustainable development.

In this study, the Life Cycle Sustainability Assessment (LCSA) proposed by Klöpfer [42] was used. LCSA is an interdisciplinary framework used for the integration of models rather than as a method itself. It is a system-based tool that considers economic, environmental, and social impacts, taking into account fundamental mechanisms. The TBL

was used as an accounting framework for the LCSA indicators, which incorporates the three pillars of sustainability through the application of the Environmental Life Cycle Assessment (LCA), Social Life Cycle Assessment (S-LCA), and Life Cycle Costing (LCC) [43]. LCA is used to analyze the environmental impacts during the life cycle of services and products. The use of this method focuses on presenting a quantitative assessment of the sustainability of buildings and has been applied in several kinds of research [16,44–47]. On the other hand, S-LCA is a relatively new field of research [48]. While most S-LCA frameworks are goal-oriented, it is possible to have some pre-defined goals (e.g., hot-spot documentation and relative goals) [49]. The assessment of social impacts proposed in this study is objective-oriented by the investment applied in the local community where the construction site is located. Finally, LCC valuation is a method of evaluating the total cost of a project or design. It makes it easier for decision makers to choose the strategy that would provide the lowest overall cost, without compromising functionality and quality [50].

Choosing between different construction materials that will provide the best economic, environmental, and social performance when applied together is a complex multicriteria decision-making (MCDM) problem, requiring interdisciplinary knowledge [28,32,51]. A multicriteria analysis is required when dealing with multi-faceted events by combining qualitative and quantitative techniques [52,53], that include the consideration of multiple criteria simultaneously [54,55]. Over the last few decades, several MCDM methods have been proposed, each with its advantages and limitations [56]. Among them, the Analytic Hierarchy Process (AHP) has the advantages of simplicity of application, repeatability, and consistency of results [57], and for that reason, it gained recognition, and currently is one of the most employed MCDM techniques by academics, and by industry in general [58–60]. The AHP method was developed by Saaty [61] and its approach is based on the fragmentation of the decision process into smaller parts, constituting a hierarchical system of objectives, metrics, and scenarios [62–64]. The methodology converts qualitative judgments into a quantitative comparison through pairwise observations between criteria and priorities, helping decision makers to reach the best possible solution to an MCDM problem [58,60,64,65].

However, despite being widely used, traditional AHP has some limitations, as its application depends directly on the prioritization of criteria assigned by experts and decision makers that can be potentially biased [65,66]. Furthermore, the use of crisp numbers derived from linguistic responses is generally not efficient in dealing with uncertainty and ambiguity, intrinsic characteristics of human thought [64,67]. Thus, conventional AHP may not reflect the true opinion of those involved in the decision-making process. This deficiency was overcome by researchers through the combination of AHP with fuzzy logic [65,68], giving rise to fuzzy AHP (FAHP). First proposed by Zadeh [69], the fuzzy set theory (FST) considers that human thought is always fuzzy when selecting one option among several others [67]. Thus, the FAHP uses fuzzy numbers instead of crisp numbers [44], converting a single vague linguistic variable into a more reliable range of variables, that is, a fuzzy number [70]. Therefore, in cases where there is great uncertainty and inconsistency, the FAHP provides more realistic results [71], minimizing the subjectivity of the process so that this characteristic does not influence the choice of the ideal solution [66]. Currently, FAHP is considered a more efficient tool than traditional AHP [72].

Relevant studies have been developed using FAHP in materials selection problems. Figueiredo et al. [44] proposed a decision-making framework using FAHP to facilitate the choice of building materials. Akadiri et al. [30] selected sustainable building materials using the Fuzzy Extended Analytic Hierarchy Process (FEAHP). Tian et al. [73] combined AHP and gray-correlation TOPSIS to select sustainable decor materials. Singh et al. [74] integrated FAHP and TOPSIS in the selection of composite materials applied to structural elements. A similar strategy was presented by Janowska-Renkas et al. [32], who used FEAHP and Fuzzy-TOPSIS to select the best high-performance concrete for post-tension bridges. Mayhoub et al. [27] applied AHP in the development of a new evaluation framework based on four green building rating systems for selecting sustainable façade materials. A similar

study was conducted by Ruslan et al. [33], who used AHP and Value-based Analysis to choose the most sustainable material for façade use, between Aluminum Composite Panel (ACP) and Stainless Steel. Dinh et al. [25] identified a list of 18 sustainability criteria applicable to the selection of sustainable materials in Vietnam, and used AHP to rank them based on their degree of relevance to the country's construction sector. Lee et al. [31] introduced a new AHP-based approach to selecting building materials based on their performance, and applied it to a case study of a concrete formwork system. Finally, Ogrodnik [75] performed a comprehensive review of MCDM methods and concluded that AHP is one of the most popular techniques currently used.

Although there is a consensus on the importance of establishing criteria and processes for the selection of suitable construction materials, few studies have been carried out to assess the sustainability of the materials considering the TBL methodology [76–78]. An even smaller number employ MCDM techniques associated with computer BIM software such as Revit and Tally [5]. Tally is a Revit plugin used to perform LCA of building materials, where materials parameterized in the BIM model are transported to an existing Life Cycle Inventory (LCI) database, collecting input and output data, including material consumption raw materials, water, energy, and the emission of solid, gaseous and liquid waste [5,79]. To fill the gap in existing research, this study proposes a BIM-LCSA-FAHP-based analysis, in order to develop a framework to select optimal building materials. In addition to the tools mentioned, this study will use the GaBi database, which is considered one of the most comprehensive tools for organizing inventory data [80] and performing impact assessments [81].

## 3. Materials and Methods

This work proposes a methodological framework that addresses different configurations of construction models in order to provide a comprehensive and systemic decision-making tool that helps to obtain the best environmental, economic, and social building materials choice when planning a new construction project.

The methodological path was developed in four steps (Figure 1).

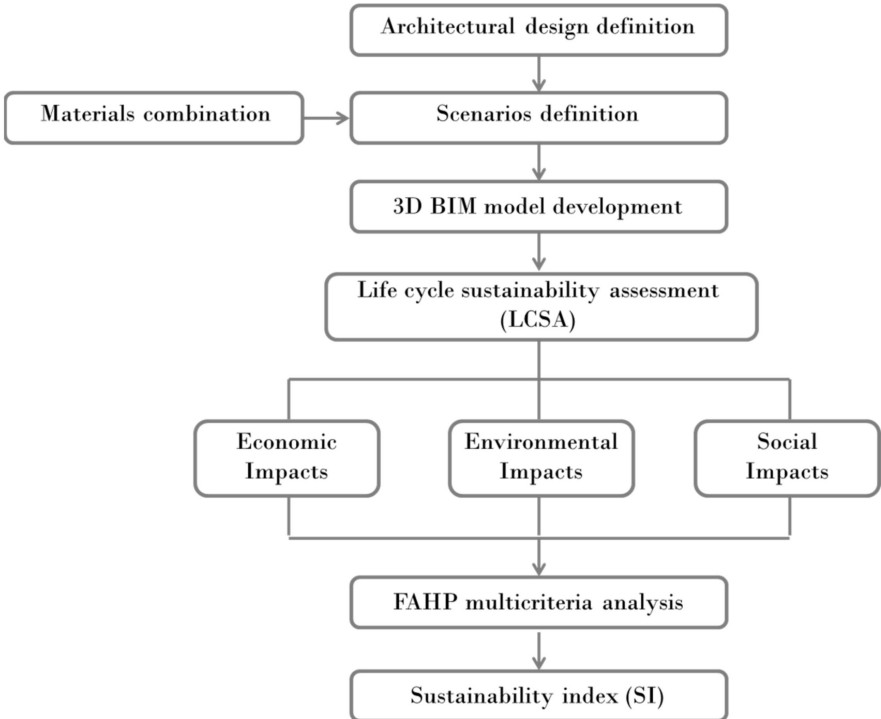

**Figure 1.** Flowchart describing the research process.

Step 1—Architectural design definition, taking into account the pattern projects characteristics for low-income buildings in Brazil.

Step 2—Selection of different building materials categories referring to the structure, roof, and painting, and their combination in different scenarios.

Step 3—Scenarios modeling in BIM software, where the characteristics of each material were inserted to parameterize the model, and analysis of the sustainability index for the created scenarios, in order to provide the necessary data to perform the multi-criteria analysis.

Step 4—Using FAHP to define which scenario gets the highest sustainability index.

In order to validate this proposal, a case study will be presented in the following sections. In using a case study to validate the proposal, we intend to present the applicability of the method and discuss how this methodological framework can be helpful to benefit the project decision-making process. In this work, the case study was chosen to be representative of the reality of Brazilian low-income construction, although the method can be applied in different contexts and cultures.

### 3.1. Architectural Design

The proposed architecture design represents the construction pattern of the most popular housing programs developed in Brazil. That is a one-story single-family residence comprising two bedrooms, living room, kitchen, and bathroom, totaling approximately 64 m² (Figure 2).

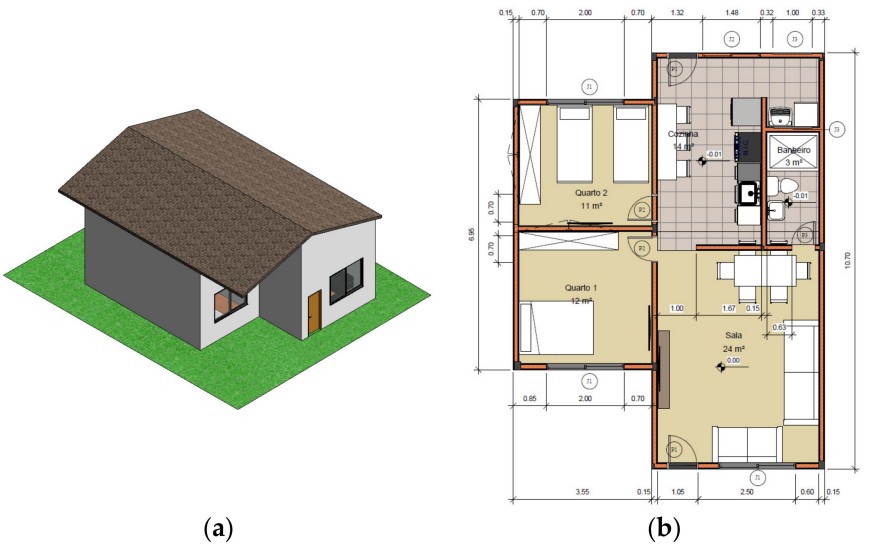

(**a**)                    (**b**)

**Figure 2.** The low-income house 3D modeling (**a**) and floor plan (**b**).

### 3.2. Materials Selection and Scenarios Definition

The materials selection considered the construction pattern of the most popular low-income government housing programs developed in Brazil, as well as its characteristics in the market, such as ease of purchase and execution. Structural materials were limited to three options with wide use in the country, namely, cast-in-place concrete, precast concrete, and structural masonry. The first is the most used due to its following technical features: (1) ease purchase of components; (2) does not require skilled labor; (3) high durability due to its resistance to shocks, vibrations, and temperature variation; and (4) high workability, that is, possibility of adapting to various forms. The second structural material was precast concrete, a constructive technology that increases the execution speed, reduces the number of workers, stocked material, and waste at the construction site. The last structural option was the structural masonry that stands out due to the use of masonry walls and stiffened slabs that act as a support structure for the buildings. Its application facilitates the rationalization of materials and greater productivity compared to conventional systems. These factors can

be lead to savings and, consequently, a more significant profit. So, it is typically used in medium or low-income residential units, where rooms are relatively small.

Regarding the painting, acrylic or PVA water-based are the most used materials, so this research was limited to these two options. Paints composed of acrylic resin are more durable, waterproof, and have greater adherence when compared to PVA water-based. In addition, the use of acrylic resin has the advantages of better pigment fixation, high gloss, good adhesion, and the fact that they are transparent and do not yellow over time. For this reason, acrylic paint is more often recommended for use outdoors due to its greater contact with moisture and deteriorating agents. Finally, regarding the roof, the options were restricted to ceramic roof tile and fiber-cement tiles. The first is one of the most used as they are low cost and have good acceptance in aesthetic terms. On the other hand, buildings that use fiber-cement tiles are quite common because of their low cost and ease of installation. Nine scenarios with different configurations of building materials were simulated, as shown in Table 1.

**Table 1.** Scenarios description.

|  | Structure | Painting | Roofing |
|---|---|---|---|
| Scenario 1 | Pre-cast concrete | PVA water-based | Ceramic |
| Scenario 2 | Pre-cast concrete | PVA water-based | Fiber cement |
| Scenario 3 | Pre-cast concrete | Acrylic | Fiber cement |
| Scenario 4 | Cast-in-place concrete | PVA water-based | Ceramic |
| Scenario 5 | Cast-in-place concrete | PVA water-based | Fiber cement |
| Scenario 6 | Cast-in-place concrete | Acrylic | Fiber cement |
| Scenario 7 | Structural Masonry | PVA water-based | Ceramic |
| Scenario 8 | Structural Masonry | PVA water-based | Fiber cement |
| Scenario 9 | Structural Masonry | Acrylic | Fiber cement |

### 3.3. BIM Modeling and Sustainability Analysis

Based on the pre-selected construction plan, the 3D BIM model was developed in Autodesk Revit, where the technical features were entered in order to parameterize each scenario with the characteristics of the corresponding materials. Considering the typology of the analyzed building, the volume of data processed during the modeling and simulation of the case study was reduced. Therefore, the computational resource used was a Core i5 notebook, with 4GB memory, and 1TB HD. However, in future studies focusing on more complex buildings, the possibility of using a cloud service should be considered, which allows access to more robust computing infrastructure. The sustainability analysis, in turn, was developed based on different methodologies (Figure 3). For the environmental impact, Tally software was employed, an environmental impact tool that applies the Autodesk Revit API to use BIM elements with a custom LCA database (GaBi). This software combines material attributes and engineering specifications with environmental impact data (GaBi) to produce LCA reports based on the information generated by the Revit model. For the calculation of the economic impact, the Brazilian Cost and Index Research System was used. This methodology is widely employed in Brazil to determine construction costs. Finally, the social impact was calculated based on the estimated amount invested in labor for the construction of a housing unit.

### 3.4. Fuzzy Analytic Hierarchy Process (FAHP) Multicriteria Analysis and Sustainability Performance

The application of the FAHP and measurement of sustainability performance can be divided into five main steps as follows.

Step 1: Formulate a pairwise comparison matrix

The first step of the FAHP is to define a scale of importance. The criteria identified in Section 3.3 were judged based on their relevance and then compared, establishing a pairwise comparison matrix, which presents the prioritization of one criterion over the other [65]. A linguistic scale was used to facilitate criteria evaluation. Finally, the linguistic

terms were converted to Triangular Fuzzy Numbers (TFN) through a fuzzy scale, leading to the formulation of a fuzzy pairwise comparison matrix [58,65]. In this study, nine TFNs were used, each represented by a membership function (Figure 4). Variables $l_{ij}$, $m_{ij}$, $u_{ij}$ represent a fuzzy number's minimum, average, and maximum values using TFN linguistic scale, as shown in Equation (1) [58].

$$\tilde{a}_{ij} = \left( l_{ij}, m_{ij}, u_{ij} \right) \tag{1}$$

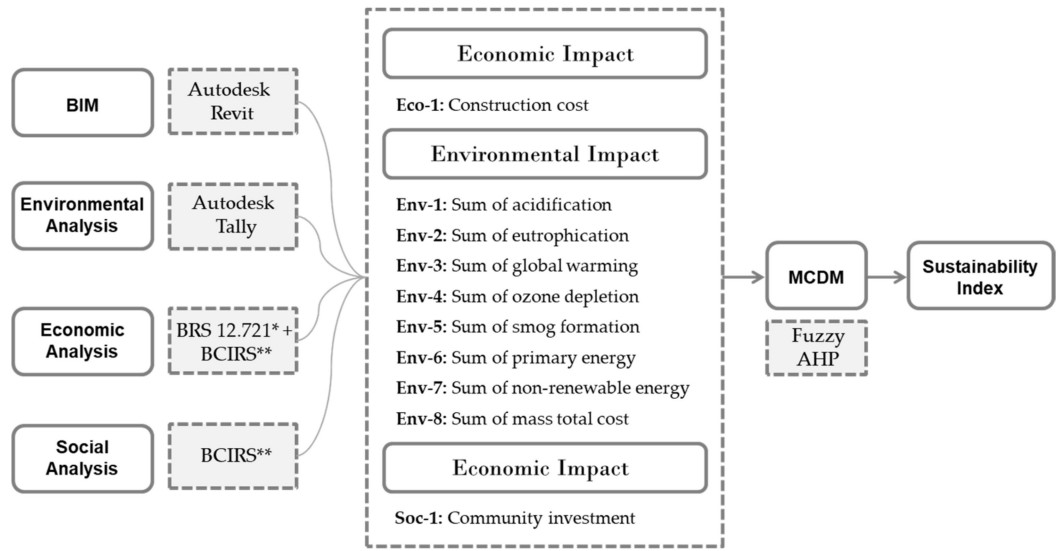

**Figure 3.** Sustainability analysis methodology; where: * BRS 12.721: Brazilian Regulatory Standard 12.721; ** BCIRS: Brazilian Cost and Index Research System.

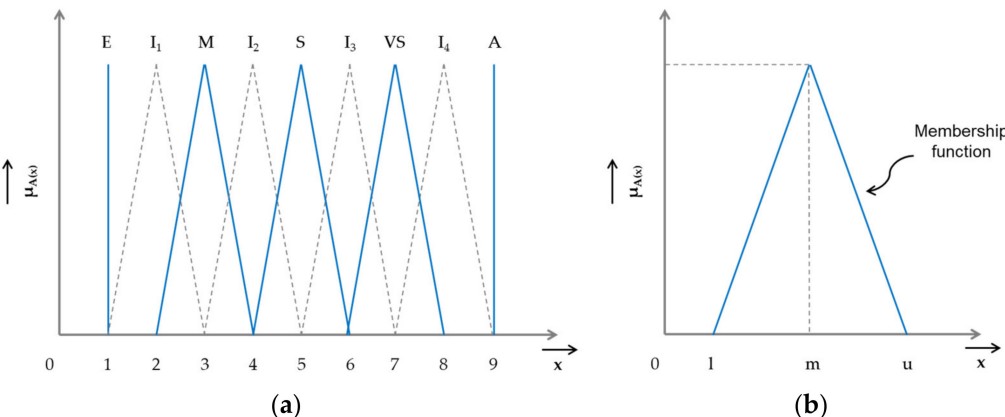

**Figure 4.** Fuzzy linguistic (**a**) fuzzy scale for FAHP with linguistic variables (**b**) geometric representation of a TFN; (Adapted from: [60,63,64,82]).

In addition, the pairwise comparison presents a reciprocal property, that is, according to Tsai and Phumchusri [63], if $\tilde{A} = (\tilde{a}_{ij}) = (l_{ij}, m_{ij}, u_{ij})$ then reciprocal value is $\tilde{A}^{-1} = (\tilde{a}^{ij})^{-1} = (l_{ij}, m_{ij}, u_{ij})^{-1} = (1/u_{ij}, 1/m_{ij}, 1/l_{ij})$ (Table 2). Thus, the elements of the comparison matrix are as shown in Equation (2):

$$\tilde{A} = (\tilde{a}_{ij}) = \begin{bmatrix} 1 & \tilde{a}_{12} & \cdots & \tilde{a}_{1n} \\ \tilde{a}_{21} & 1 & \cdots & \tilde{a}_{2n} \\ \vdots & \vdots & \ddots & \vdots \\ \tilde{a}_{n1} & \tilde{a}_{n2} & \cdots & 1 \end{bmatrix} \tag{2}$$

where: $\tilde{a}_{ij} = 1$, if $i = j$; $\tilde{a}_{ij} = (\tilde{a}_{ij})^{-1}$, if $i \neq j$.

**Table 2.** Linguistic terms and TFN reciprocal scale.

| Linguistic Scale | Evaluation Criterion | Triangular Fuzzy Number (TFN) | TFN Reciprocal Scale |
|---|---|---|---|
| Equally Important (E) | 1 | (1,1,1) | (1,1,1) |
| Intermediate Value ($I_1$) | 2 | (1,2,3) | (1/3,1/2,1) |
| Moderately Important (M) | 3 | (2,3,4) | (1/4,1/3,1/2) |
| Intermediate Value ($I_2$) | 4 | (3,4,5) | (1/5,1/4,1/3) |
| Strongly Important (S) | 5 | (4,5,6) | (1/6,1/5,1/4) |
| Intermediate Value ($I_3$) | 6 | (5,6,7) | (1/7,1/6,1/5) |
| Very Strongly Important (VS) | 7 | (6,7,8) | (1/8,1/7,1/6) |
| Intermediate Value ($I_4$) | 8 | (7,8,9) | (1/9,1/8,1/7) |
| Absolutely Important (A) | 9 | (9,9,9) | (1/9,1/9,1/9) |

Adapted from: [57,58,83,84].

Step 2: Compute fuzzy weights using geometric mean method

The fuzzy mean geometric value of the ith criterion against each criterion was determined using the geometric mean method, computing the fuzzy triangular vector $r_i$ Equation (3). The relative fuzzy criteria weight of the ith criterion is represented as a TFN and computed applying Equation (4) [64–66,82].

$$r_i = \left( \tilde{a}_{i1} \otimes \ldots \otimes \tilde{a}_{ij} \otimes \ldots \otimes \tilde{a}_{in} \right)^{1/n} \tag{3}$$

$$\omega_i = r_i \otimes \left[ r_1 \oplus \ldots \oplus r_i \oplus \ldots \oplus r_n \right]^{-1} = (l_{\omega i}, m_{\omega i}, u_{\omega i}) \tag{4}$$

where:

$\tilde{a}_{ij}$ is the comparison of fuzzy value from criterion $i$ to $j$;

$r_i$ is the geometric mean value for comparison of the fuzzy value of criterion i to the other criterion;

$\omega_i$ is the fuzzy weight of each criterion;

$n$ is the number of criteria.

Step 3: Defuzzify criteria weights using center of area method

The fuzzy weights $\omega_i$ previously calculated are still TFNs; therefore, average weights of the criteria must be calculated and defuzzified through the center of area approach, applying Equation (5) [58,66]. Defuzzification was used to convert the fuzzy values to exact values [63]. Finally, the normalized criteria weights were calculated using Equation (6) [58].

$$M_i = \frac{l_{\omega i} \oplus m_{\omega i} \oplus u_{\omega i}}{3} \tag{5}$$

$$W_i = \frac{M_i}{\sum_{i=1}^{n} M_i} \tag{6}$$

Step 4: Consistency check

The judgment matrix obtained through the FAHP technique is not always plausible [59]. Therefore, a consistency test was performed for the pairwise comparison matrix ($\tilde{A}$). First, the maximum eigenvalue ($\lambda_{max}$) was computed using Equation (7) [58]. Then, the Consistency Index (CI) and the Consistency Ratio (CR) were calculated applying Equations (8) and (9) [85].

$$\lambda_{max} = \sum_{i=1}^{n} \frac{\sum_{j=1}^{a} a_{ij} \times \omega_i}{\omega_i} ; i = 1, 2, 3, \ldots, n \text{ and } j = 1, 2, 3, \ldots, n \tag{7}$$

$$CI = \frac{\lambda_{max} - n}{n - 1} \tag{8}$$

$$CR = \frac{CI}{RI} \tag{9}$$

where:

$n$ is the order of the comparison matrix, that is, the number of criteria;

$RI$ is the Random Matrix Index that can be obtained from Saaty [61], based on the number of criteria used during the assessment.

If $CR < 0.10$, the degree of consistency is considered satisfactory. Otherwise, evaluations are inconsistent, and the rating $a_{ij}$ in the pairwise comparison matrix must be revised to remove inconsistency [57,59,63,66,85,86].

Step 5: Determine performance score

Finally, based on the weights determined for each TBL sub-criterion through the FAHP, and on the values found in the Life Cycle Sustainability Assessment (LCSA), it is possible to determine the Sustainability Index (*SI*) using Equation (10) [87].

$$SI = \sum_{n=1}^{n} W_i \times X_i \tag{10}$$

where:

$W_i$ is the calculated weight for each criterion;

$X_i$ is the calculated impact for each criterion of each scenario.

## 4. Results

### 4.1. Economic Impact

The economic impact assessment was based on the Brazilian Regulatory Standard 12.721, entitled "Evaluation for unit costs and elaborations of construction budget for incorporation of joint ownership building—Procedure". For low-income buildings, such as the one used in this case study, the aforementioned standard establishes an approximate cost of USD 334.36/built m$^2$ (November 2021 values). That is, the total construction cost, including projects and construction, was in the order of USD 28,645.64. For each modeled scenario, the Brazilian Cost and Index Research System was used to identify the cost variations related to the constructive variations presented (Table 3).

**Table 3.** Scenarios global cost.

| Scenario | 1 | 2 | 3 | 4 | 5 | 6 | 7 | 8 | 9 |
|---|---|---|---|---|---|---|---|---|---|
| Cost (USD) | 26,290.95 | 26,308.86 | 27,651.14 | 28,645.64 | 28,663.55 | 30,005.83 | 29,294.15 | 29,312.07 | 30,654.35 |

### 4.2. Environmental Impact

The environmental impact analysis followed stages A, B and C, which are characterized as the "cradle to grave", and represented by Figure 5. Stage D adopted by the GaBi database, as a reuse and recycling stage of the building, was removed from this analysis since the parameters used in the international standard differ from the model adopted in the Brazilian market [88].

Based on the previously built parameterized BIM model, Tally software was employed to perform the LCA analysis for each scenario, using a custom database (GaBi). The analysis was carried out separately for phases A, B and C, in order to highlight the strengths and weaknesses of each scenario regarding the building's environmental impact degree in each phase of its life cycle. Lastly, the simulation was performed considering the entire lifespan of the building, that is, all of the steps involved were combined, illustrating the scenario that has the least impact on the environment.

The results of Table 4 are related to steps A1 to A4, that is, materials extraction and transport stage. The scenarios that used the structural masonry system stood out negatively, with the highest environmental impact results, in 8 of the 9 metrics analyzed. The precast concrete structure, on the other hand, causes the least environmental impacts at this stage.

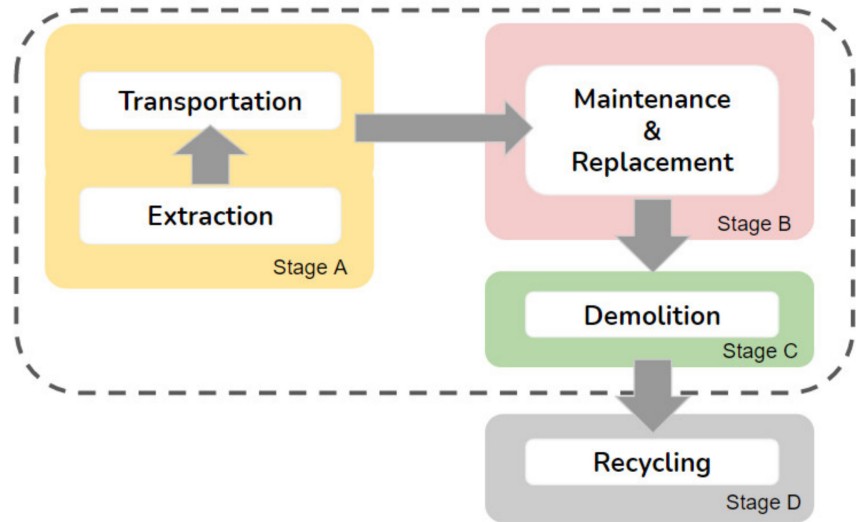

**Figure 5.** Environmental impact analysis system boundary.

**Table 4.** Environmental impact analysis results for stages A1 to A4.

| Scenario | Environmental Metrics | | | | | | | |
|---|---|---|---|---|---|---|---|---|
| | Env-1 (kgSO$_2$eq) | Env-2 (kgNeq) | Env-3 (kgCO$_2$eq) | Env-4 (CFC-11qe) | Env-5 (kgO$_3$eq) | Env-6 (MJ) | Env-7 (MJ) | Env-8 (kg) |
| 1 | $5.98 \times 10^6$ | $3.81 \times 10^5$ | $4.80 \times 10^9$ | $9.96 \times 10^{-6}$ | $1.22 \times 10^8$ | $4.51 \times 10^{10}$ | $4.24 \times 10^{10}$ | $2.72 \times 10^9$ |
| 2 | $5.98 \times 10^6$ | $3.88 \times 10^5$ | $4.76 \times 10^9$ | $2.68 \times 10^{-6}$ | $1.23 \times 10^8$ | $4.37 \times 10^{10}$ | $4.09 \times 10^{10}$ | $2.75 \times 10^9$ |
| 3 | $6.28 \times 10^6$ | $3.85 \times 10^5$ | $4.75 \times 10^9$ | $4.45 \times 10^{-6}$ | $1.20 \times 10^8$ | $4.35 \times 10^{10}$ | $4.08 \times 10^{10}$ | $2.71 \times 10^9$ |
| 4 | $6.12 \times 10^6$ | $3.84 \times 10^5$ | $4.82 \times 10^9$ | $8.38 \times 10^{-6}$ | $1.24 \times 10^8$ | $4.55 \times 10^{10}$ | $4.28 \times 10^{10}$ | $2.73 \times 10^9$ |
| 5 | $6.11 \times 10^6$ | $3.91 \times 10^5$ | $4.78 \times 10^9$ | $1.09 \times 10^{-6}$ | $1.24 \times 10^8$ | $4.40 \times 10^{10}$ | $4.13 \times 10^{10}$ | $2.76 \times 10^9$ |
| 6 | $6.41 \times 10^6$ | $3.88 \times 10^5$ | $4.77 \times 10^9$ | $2.86 \times 10^{-6}$ | $1.22 \times 10^8$ | $4.39 \times 10^{10}$ | $4.11 \times 10^{10}$ | $2.71 \times 10^9$ |
| 7 | $1.02 \times 10^7$ | $6.92 \times 10^5$ | $6.13 \times 10^9$ | $-5.75 \times 10^{-6}$ | $1.98 \times 10^8$ | $6.59 \times 10^{10}$ | $6.16 \times 10^{10}$ | $4.22 \times 10^9$ |
| 8 | $1.02 \times 10^7$ | $7.00 \times 10^5$ | $6.09 \times 10^9$ | $-1.30 \times 10^{-8}$ | $1.99 \times 10^8$ | $6.44 \times 10^{10}$ | $6.02 \times 10^{10}$ | $4.24 \times 10^9$ |
| 9 | $1.04 \times 10^7$ | $6.96 \times 10^5$ | $6.09 \times 10^9$ | $-1.15 \times 10^{-8}$ | $1.97 \times 10^8$ | $6.43 \times 10^{10}$ | $6.00 \times 10^{10}$ | $4.20 \times 10^9$ |

The second stage, illustrated in Table 5, shows the environmental impact analysis results for stages B2 to B5, representing the maintenance and replacement of materials. It is possible to observe an inversion of scenarios, whereby the use of structural masonry generated the least impact, with 8 out of 9 results. At this stage, cast concrete appears to have the most significant impact in all analyzed metrics.

**Table 5.** Environmental impact analysis results for stages B2 to B5.

| Scenario | Environmental Metrics | | | | | | | |
|---|---|---|---|---|---|---|---|---|
| | Env-1 (kgSO$_2$eq) | Env-2 (kgNeq) | Env-3 (kgCO$_2$eq) | Env-4 (CFC-11qe) | Env-5 (kgO$_3$eq) | Env-6 (MJ) | Env-7 (MJ) | Env-8 (kg) |
| 1 | $1.82 \times 10^6$ | $1.42 \times 10^5$ | $4.15 \times 10^8$ | $1.94 \times 10^{-8}$ | $4.01 \times 10^7$ | $8.74 \times 10^9$ | $7.44 \times 10^9$ | $1.30 \times 10^9$ |
| 2 | $1.88 \times 10^6$ | $1.45 \times 10^5$ | $4.11 \times 10^8$ | $-9.03 \times 10^{-10}$ | $4.18 \times 10^7$ | $8.70 \times 10^9$ | $7.37 \times 10^9$ | $1.34 \times 10^9$ |
| 3 | $1.77 \times 10^6$ | $6.27 \times 10^4$ | $1.58 \times 10^8$ | $5.31 \times 10^{-6}$ | $1.09 \times 10^7$ | $3.47 \times 10^9$ | $2.76 \times 10^9$ | $7.06 \times 10^8$ |
| 4 | $1.82 \times 10^6$ | $1.42 \times 10^5$ | $4.15 \times 10^8$ | $1.94 \times 10^{-8}$ | $4.01 \times 10^7$ | $8.74 \times 10^9$ | $7.44 \times 10^9$ | $1.30 \times 10^9$ |
| 5 | $1.88 \times 10^6$ | $1.45 \times 10^5$ | $4.11 \times 10^8$ | $-9.03 \times 10^{-10}$ | $4.18 \times 10^7$ | $8.70 \times 10^9$ | $7.37 \times 10^9$ | $1.34 \times 10^9$ |
| 6 | $1.77 \times 10^6$ | $6.27 \times 10^4$ | $1.58 \times 10^8$ | $5.31 \times 10^{-6}$ | $1.09 \times 10^7$ | $3.47 \times 10^9$ | $2.76 \times 10^9$ | $7.06 \times 10^8$ |
| 7 | $1.61 \times 10^6$ | $1.25 \times 10^5$ | $3.63 \times 10^8$ | $1.94 \times 10^{-8}$ | $3.49 \times 10^7$ | $7.70 \times 10^9$ | $6.52 \times 10^9$ | $1.19 \times 10^9$ |
| 8 | $1.66 \times 10^6$ | $1.28 \times 10^5$ | $3.59 \times 10^8$ | $2.64 \times 10^{-12}$ | $3.66 \times 10^7$ | $7.67 \times 10^9$ | $6.45 \times 10^9$ | $1.23 \times 10^9$ |
| 9 | $1.58 \times 10^6$ | $5.77 \times 10^4$ | $1.43 \times 10^8$ | $4.54 \times 10^{-6}$ | $1.03 \times 10^7$ | $3.20 \times 10^9$ | $2.51 \times 10^9$ | $6.84 \times 10^8$ |

The last stage is C2 to C4, presented in Table 6, which consists of the demolition or disposal of the house. Precast concrete is once again the lowest impact option in seven out of eight metrics. Structural masonry is again the option with a greater impact, occupying seven of the eight positions.

**Table 6.** Environmental impact analysis results for stages C2 to C4.

| Scenario | Environmental Metrics | | | | | | | |
|---|---|---|---|---|---|---|---|---|
| | Env-1 (kgSO$_2$eq) | Env-2 (kgNeq) | Env-3 (kgCO$_2$eq) | Env-4 (CFC-11qe) | Env-5 (kgO$_3$eq) | Env-6 (MJ) | Env-7 (MJ) | Env-8 (kg) |
| 1 | $1.26 \times 10^6$ | $7.70 \times 10^4$ | $2.73 \times 10^8$ | $8.58 \times 10^{-10}$ | $2.40 \times 10^7$ | $4.45 \times 10^9$ | $4.16 \times 10^9$ | - |
| 2 | $1.24 \times 10^6$ | $7.61 \times 10^4$ | $2.69 \times 10^8$ | $8.51 \times 10^{-10}$ | $2.37 \times 10^7$ | $4.39 \times 10^9$ | $4.10 \times 10^9$ | - |
| 3 | $1.24 \times 10^6$ | $7.71 \times 10^4$ | $2.69 \times 10^8$ | $8.52 \times 10^{-10}$ | $2.37 \times 10^7$ | $4.39 \times 10^9$ | $4.10 \times 10^9$ | - |
| 4 | $1.26 \times 10^6$ | $7.71 \times 10^4$ | $2.73 \times 10^8$ | $8.59 \times 10^{-10}$ | $2.41 \times 10^7$ | $4.45 \times 10^9$ | $4.17 \times 10^9$ | - |
| 5 | $1.24 \times 10^6$ | $7.62 \times 10^4$ | $2.69 \times 10^8$ | $8.52 \times 10^{-10}$ | $2.37 \times 10^7$ | $4.39 \times 10^9$ | $4.11 \times 10^9$ | - |
| 6 | $1.24 \times 10^6$ | $7.72 \times 10^4$ | $2.70 \times 10^8$ | $8.52 \times 10^{-10}$ | $2.37 \times 10^7$ | $4.39 \times 10^9$ | $4.11 \times 10^9$ | - |
| 7 | $2.02 \times 10^6$ | $1.14 \times 10^5$ | $4.39 \times 10^8$ | $8.52 \times 10^{-10}$ | $3.92 \times 10^7$ | $7.26 \times 10^9$ | $6.79 \times 10^9$ | - |
| 8 | $2.00 \times 10^6$ | $1.13 \times 10^5$ | $4.36 \times 10^8$ | $8.45 \times 10^{-10}$ | $3.88 \times 10^7$ | $7.20 \times 10^9$ | $6.73 \times 10^9$ | - |
| 9 | $2.00 \times 10^6$ | $1.14 \times 10^5$ | $4.36 \times 10^8$ | $8.45 \times 10^{-10}$ | $3.88 \times 10^7$ | $7.20 \times 10^9$ | $6.74 \times 10^9$ | - |

Finally, all the scenarios were combined to generate the environmental impact analysis, illustrated in Table 7. This cradle-to-grave analysis highlighted the lesser impact caused by the use of precast concrete (Scenarios 1 to 3). Structural masonry presented greater impact in eight of the nine metrics (Scenarios 7 to 9).

**Table 7.** Environmental impact analysis results for all stages, A1 to C4.

| Scenario | Environmental Metrics | | | | | | | |
|---|---|---|---|---|---|---|---|---|
| | Env-1 (kgSO$_2$eq) | Env-2 (kgNeq) | Env-3 (kgCO$_2$eq) | Env-4 (CFC-11qe) | Env-5 (kgO$_3$eq) | Env-6 (MJ) | Env-7 (MJ) | Env-8 (kg) |
| 1 | $9.16 \times 10^6$ | $6.08 \times 10^5$ | $5.55 \times 10^9$ | $2.50 \times 10^{-8}$ | $1.91 \times 10^8$ | $5.87 \times 10^{10}$ | $5.45 \times 10^{10}$ | $4.13 \times 10^9$ |
| 2 | $9.17 \times 10^6$ | $6.16 \times 10^5$ | $5.50 \times 10^9$ | $-1.83 \times 10^{-6}$ | $1.92 \times 10^8$ | $5.71 \times 10^{10}$ | $5.29 \times 10^{10}$ | $4.19 \times 10^9$ |
| 3 | $9.37 \times 10^6$ | $5.32 \times 10^5$ | $5.24 \times 10^9$ | $5.26 \times 10^{-6}$ | $1.59 \times 10^8$ | $5.17 \times 10^{10}$ | $4.28 \times 10^{10}$ | $3.51 \times 10^9$ |
| 4 | $9.28 \times 10^6$ | $6.10 \times 10^5$ | $5.56 \times 10^9$ | $2.39 \times 10^{-8}$ | $1.92 \times 10^8$ | $5.90 \times 10^{10}$ | $5.48 \times 10^{10}$ | $4.14 \times 10^9$ |
| 5 | $9.30 \times 10^6$ | $6.18 \times 10^5$ | $5.51 \times 10^9$ | $-2.88 \times 10^{-6}$ | $1.93 \times 10^8$ | $5.74 \times 10^{10}$ | $5.32 \times 10^{10}$ | $4.20 \times 10^9$ |
| 6 | $9.50 \times 10^6$ | $5.34 \times 10^5$ | $5.25 \times 10^9$ | $4.20 \times 10^{-6}$ | $1.60 \times 10^8$ | $5.20 \times 10^{10}$ | $4.85 \times 10^{10}$ | $3.52 \times 10^9$ |
| 7 | $1.37 \times 10^7$ | $9.30 \times 10^5$ | $6.85 \times 10^9$ | $2.04 \times 10^{-8}$ | $2.75 \times 10^8$ | $7.99 \times 10^{10}$ | $7.41 \times 10^{10}$ | $5.76 \times 10^9$ |
| 8 | $1.37 \times 10^7$ | $9.38 \times 10^5$ | $6.80 \times 10^9$ | $-6.40 \times 10^{-6}$ | $2.76 \times 10^8$ | $7.83 \times 10^{10}$ | $7.25 \times 10^{10}$ | $5.82 \times 10^9$ |
| 9 | $1.38 \times 10^7$ | $8.66 \times 10^5$ | $6.57 \times 10^9$ | $-3.50 \times 10^{-7}$ | $2.48 \times 10^8$ | $7.37 \times 10^{10}$ | $6.84 \times 10^{10}$ | $5.24 \times 10^9$ |

*4.3. Social Impact*

Considering that the construction industry is of great importance in the generation of jobs, and annually moves significant financial resources [4], the interpretation of its social impact in a community must be evaluated according to the volume of locally invested resources during the execution of the work. Given the typology employed in this case study, and based on the Brazilian Cost and Index Research System, of the total amount invested in the construction of a single unit, 53.33% of the financial resources are directed to labor. Thus, if the labor is hired locally, 53.33% of the total cost will have a social impact on the community. Therefore, through the total values calculated for each analyzed scenario (Table 3), it is possible to estimate the social impact provided for each one of them (Table 8).

**Table 8.** Community social impact.

| Scenario | 1 | 2 | 3 | 4 | 5 | 6 | 7 | 8 | 9 |
|---|---|---|---|---|---|---|---|---|---|
| Social Impact USD | 14,020.96 | 14,030.52 | 14,746.36 | 15,259.90 | 15,286.27 | 16,002.11 | 15,622.57 | 15,632.13 | 16,347.97 |

*4.4. FAHP Multicriteria Analysis*

The pairwise comparison matrix was established by comparing one criterion over the other. In this study, the economic impact had the greatest relevance, since a low-income building project needs to adapt to the purchase possibilities of future users. The second level of importance was the social impact, which will affect the entire community surrounding the construction. Finally, for the environmental area, eight variables were studied, allowing for a more detailed analysis of the LCSA. Each environmental metric received the same level of importance. The results of the scale of importance distribution, converted to the TFN reciprocal scale, are illustrated in Table 9.

**Table 9.** Pairwise comparison fuzzy matrix converted to TFN reciprocal scale.

| Metric | Eco-1 | Env-1 | Env-2 | Env-3 | Env-4 | Env-5 | Env-6 | Env-7 | Env-8 | Soc-1 |
|--------|-------|-------|-------|-------|-------|-------|-------|-------|-------|-------|
| Eco-1 | (1,1,1) | (6,7,8) | (6,7,8) | (6,7,8) | (6,7,8) | (6,7,8) | (6,7,8) | (6,7,8) | (6,7,8) | (2,3,4) |
| Env-1 | (1/8,1/7,1/6) | (1,1,1) | (1,1,1) | (1,1,1) | (1,1,1) | (1,1,1) | (1,1,1) | (1,1,1) | (1,1,1) | (1/4,1/3,1/2) |
| Env-2 | (1/8,1/7,1/6) | (1,1,1) | (1,1,1) | (1,1,1) | (1,1,1) | (1,1,1) | (1,1,1) | (1,1,1) | (1,1,1) | (1/4,1/3,1/2) |
| Env-3 | (1/8,1/7,1/6) | (1,1,1) | (1,1,1) | (1,1,1) | (1,1,1) | (1,1,1) | (1,1,1) | (1,1,1) | (1,1,1) | (1/4,1/3,1/2) |
| Env-4 | (1/8,1/7,1/6) | (1,1,1) | (1,1,1) | (1,1,1) | (1,1,1) | (1,1,1) | (1,1,1) | (1,1,1) | (1,1,1) | (1/4,1/3,1/2) |
| Env-5 | (1/8,1/7,1/6) | (1,1,1) | (1,1,1) | (1,1,1) | (1,1,1) | (1,1,1) | (1,1,1) | (1,1,1) | (1,1,1) | (1/4,1/3,1/2) |
| Env-6 | (1/8,1/7,1/6) | (1,1,1) | (1,1,1) | (1,1,1) | (1,1,1) | (1,1,1) | (1,1,1) | (1,1,1) | (1,1,1) | (1/4,1/3,1/2) |
| Env-7 | (1/8,1/7,1/6) | (1,1,1) | (1,1,1) | (1,1,1) | (1,1,1) | (1,1,1) | (1,1,1) | (1,1,1) | (1,1,1) | (1/4,1/3,1/2) |
| Env-8 | (1/8,1/7,1/6) | (1,1,1) | (1,1,1) | (1,1,1) | (1,1,1) | (1,1,1) | (1,1,1) | (1,1,1) | (1,1,1) | (1/4,1/3,1/2) |
| Soc-1 | (1/4,1/3,1/2) | (2,3,4) | (2,3,4) | (2,3,4) | (2,3,4) | (2,3,4) | (2,3,4) | (2,3,4) | (2,3,4) | (1,1,1) |

Table 10 presents the relative fuzzy weight for each criterion represented as a TFN, as well as the single normalized criteria weights after defuzzification, calculated from the weights adopted for each metric in the step of measuring its relevance.

**Table 10.** Matrix of fuzzy weights.

| Criteria | Metric | Fuzzy Weight | | | Metric Weight | Criteria Weight |
|----------|--------|-----|-----|-----|---------------|-----------------|
| | | *l* | *m* | *u* | | |
| Economic | Eco-1: Cost | 0.385 | 0.397 | 0.401 | 39.41% | 39.41% |
| Environmental | Env-1: Sum of acidification | 0.052 | 0.055 | 0.061 | 5.58% | 44.64% |
| | Env-2: Sum of eutrophication | 0.052 | 0.055 | 0.061 | 5.58% | |
| | Env-3: Sum of global warming | 0.052 | 0.055 | 0.061 | 5.58% | |
| | Env-4: Sum of ozone depletion | 0.052 | 0.055 | 0.061 | 5.58% | |
| | Env-5: Sum of smog formation | 0.052 | 0.055 | 0.061 | 5.58% | |
| | Env-6: Sum of primary energy | 0.052 | 0.055 | 0.061 | 5.58% | |
| | Env-7: Sum of non-renewable energy | 0.052 | 0.055 | 0.061 | 5.58% | |
| | Env-8: Sum of mass total cost | 0.052 | 0.055 | 0.061 | 5.58% | |
| Social | Soc-1: Community investment | 0.130 | 0.162 | 0.187 | 15.95% | 15.95% |

The consistency test was performed using Equations (7) to (9), which validated the adopted criteria weights, presenting the results shown in Equation (11). As the calculated *CR* was below 0.10, the degree of consistency was considered satisfactory.

$$\lambda_{max} = 10,005 \ \therefore \ CI = 5.632 \times 10^{-4} \ \therefore \ CR = 3.780 \times 10^{-4} \tag{11}$$

Finally, the performance score of each criterion was calculated, allowing for the elaboration of the Sustainability Index (*SI*) presented in Table 11.

**Table 11.** Sustainability Index (*SI*).

| Position (↑ Sustainable) | Scenario | *SI* |
|:---:|:---:|:---:|
| 1st | 3 | 0.091 |
| 2nd | 6 | 0.058 |
| 3rd | 1 | −0.062 |
| 4th | 4 | −0.065 |
| 5th | 7 | −0.082 |
| 6th | 9 | −0.091 |
| 7th | 2 | −0.116 |
| 8th | 5 | −0.148 |
| 9th | 8 | −0.267 |

## 5. Discussion

The application of the proposed method, integrating LCSA, BIM and MCDM, demonstrated that Scenario 3 was considered the most sustainable option for the analyzed building. It is interesting to highlight that among the three categories of materials analyzed, those that presented the best results for this case study were precast concrete (structural material), acrylic paint (painting), and fiber-cement tiles (roofing). Identifying and modeling the implications of individual choices has always been fundamental to building construction. In this context, Scenario 3 was ranked the third lowest in terms of cost, had the lowest environmental impact, and the third-highest positive social impact, making this scenario the best option for this type of low-income housing unit.

It is essential to point out that the methodological framework proposed in this study was tested for a proposed building. This is valid since many publications in the literature present similar validations, but it brings certain limitations to the study compared to an actual building. For example, regarding the economic criterion analyzed, information from the Brazilian Cost and Index Research System was considered instead of collecting data in the field (i.e., price surveys in stores in the construction region). This was also the case for the social criterion since the real impact of a building of this size on society was not investigated in the region. In an actual project, the weights of the criteria could be decided, for example, through questionnaires made available to all those directly or indirectly involved in the building construction and operation.

Besides, when applied to real case studies, impact categories should be chosen according to stakeholder needs and priorities. Environmental impact categories, for example, can be chosen in terms of the most critical factors in the region. However, this work aims to show the importance and applicability of considering the three pillars of sustainability during any building assessment. Even though the weights of the criteria can vary drastically from one project to another, the TBL approach must be applied to ensure the development of more sustainable construction industry.

In this study, the primary aim was to present a methodological path that could be replicated for any type of construction, considering the reality of any country. Therefore, the authors understand that the objective of the analysis has been achieved by presenting a proposal that different professionals can apply to improve the decision-making process of which building materials are more sustainable in each case. Finally, it is worth mentioning that the case study presented focuses on the construction of low-income buildings, which is a subject that still lacks much discussion in the literature. With the proposed methodological framework, it is clear that it is possible to consider the three pillars of sustainability together, even when the project budget is limited.

## 6. Conclusions

A framework was developed to integrate LCSA, MCDM, and BIM, to assess the most sustainable choice of materials applied to the construction of low-income buildings, according to the three pillars of the Triple Bottom Line (TBL). Initially, a 3D BIM model was

developed. Next, an environmental impact analysis was performed using Tally, an Autodesk Revit plugin that uses the Gabi database. Brazilian Regulatory Standard 12.721 and the Brazilian System for Research on Costs and Indexes were used to perform the economic and social impact analysis. The economic, environmental, and social criteria were considered separately, and then together, making it possible to determine the most sustainable option for this building typology. The proposed methodology was tested for a building for which the construction pattern is one of the most popular in low-income housing projects developed in Brazil. In this case study, nine scenarios were simulated, representing a combination of the following three different categories of materials: Structural (cast-in-place concrete, precast concrete, and structural masonry); Painting (acrylic paint and PVA water-based); and roofing (ceramic roof tile and fiber-cement tiles). The results obtained separately for each criterion in the BIM-LCSA analysis were submitted to a multicriteria analysis technique (FAHP) to identify the most advantageous scenario. It should be noted that the same scope is replicable for other build typologies, increasing the impact of the database. Regarding the best option for low-income construction, the precast concrete, when combined with acrylic paint and fiber cement tiles (scenario 3), proved to be the most advantageous considering all the TBL's three pillars, and achieved first place in the sustainability index (*SI*) developed in this work.

Although the application of this framework can contribute to increased efficiency in the execution of low-income works, this research is subject to some limitations that should be considered, and some may serve as a stimulus for future work. First, this study does not cover other construction systems such as sewage, electrical and hydraulic systems. Measuring the impacts obtained at all stages of construction and comparing the impacts generated with the parameters obtained in this work constitute the next steps of this research. Second, the environmental analysis of reuse and recycling (stage D) was not included in this study, since the international standard presents different characteristics in the process of selecting building materials. So, this research used the Gabi database pattern, which is included in the Tally plugin. Third, there are limitations regarding the social indicator used, as it is only related to the direct impact on the local community where the building is being constructed. Therefore, the extension of social indicators in the proposed framework is required in future work. In addition, there is no national database that allows for a more reliable decision-making process based on the characteristics of local materials. Thus, the creation of national and international databases is urgent and necessary, to facilitate a more conscious decision-making process with regard to materials. Fourth, this case study focused on the analysis of a social housing, where the volume of modeled information is limited. However, conducting research in more complex buildings may require access to advanced computing infrastructure. In this sense, collaborations with research projects that develop cloud-based platform services, such as the IOTWINS project, can improve the adoption and applicability of the proposed framework [89]. Finally, the current research can be extended in several directions to bring the methodology closer to the real-life materials-choice process. The use of new material options such as walls, cladding, flooring and window frames, among others, should be incorporated in future work.

**Author Contributions:** Conceptualization, A.N.H. and M.V.A.P.M.F.; methodology, M.V.A.P.M.F., M.N., K.V.F. and B.B.F.d.C.; software, M.V.A.P.M.F., M.N. and K.V.F.; validation, M.V.A.P.M.F., A.N.H. and M.B.d.M.; formal analysis, M.V.A.P.M.F., A.N.H. and M.B.d.M.; investigation, M.V.A.P.M.F., M.N., K.V.F. and B.B.F.d.C.; resources, M.V.A.P.M.F., M.N., K.V.F. and A.N.H.; data curation, M.V.A.P.M.F., M.N., K.V.F. and B.B.F.d.C.; writing—original draft preparation, M.V.A.P.M.F., M.N., A.N.H. and B.B.F.d.C.; writing—review and editing, B.B.F.d.C. and A.N.H.; visualization, B.B.F.d.C. and A.N.H.; supervision, A.N.H.; project administration, A.N.H. All authors have read and agreed to the published version of the manuscript.

**Funding:** This research received no external funding.

**Institutional Review Board Statement:** Not applicable.

**Informed Consent Statement:** Not applicable.

**Data Availability Statement:** Data will be available upon reasonable request.

**Acknowledgments:** Assed Haddad would like to acknowledge CNPq (Conselho Nacional de Desenvolvimento Científico e Tecnológico), Brasilia, DF, Brazil (the Brazilian National Research Council), and Fundação Carlos Chagas Filho de Amparo à Pesquisa do Estado do Rio de Janeiro (FAPERJ) supporting Brazilian researches.

**Conflicts of Interest:** The authors declare no conflict of interest.

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
