# Peer review of "Sustainability Assessment of a Low-Income Building: A BIM-LCSA-FAHP-Based Analysis"

_buildings, doi:10.3390/buildings12020181_

Round 1
Reviewer 1 Report
The manuscript describe the definition of a novel framework which adopts a set of computing models to tackle different problems coming from construction industry.
The study performed by the authors is relevant and the effort made is appreciated.
The manuscript shows how they tackle the problem of producing new buildings, but without further burdening the environment.
To do that, they adopt a defined set of computing models through a new framework which has been applyed and tested on nine scenarios, addressing different configurations of construction models.
The aim of the present study is to provide a comprehensive and systemic decision-making tool that enables the best environmental, economic, and social building materials choice when planning a new construction project.
The method defined, the models adopted, together with their application into the different usecase and the related impact analysis are well explained.
The paper, even if well written, lack of some information that have to be added.
(i) A clear description of the computatinal resources used to test the model is missing. Today, the computing infrastructure (or computing resources and related services) for such kind of research is becaming more and more important and there is no mention in the manuscript about the resources and the services used to reach the final goal. As an example, a growing number of such activities is making use of cloud services (made available within funded projects as well as private companies).
The last sentence introduce to another point missing in the manuscript which is (ii) the collaboration and interaction with other group(s) or project(s), specially with those that adopt the same approach and/or develop computing-based services.
To this end, to complete the information provided by the manuscript, a reference to some collaborations and/or EU funded projects running to date can be added in terms of futhure or possible collaborations.
Among the projects to consider, the IOTWINS project (https://cordis.europa.eu/project/id/857191/reporting) is aimed to provide cloud-based tools and services for (but not limeted to) the hybrid digital twins. In particular, they design a cloud-based platform services that can be exported adn implemented to other computing models.
More information is described in the following publication that should be cited - e.g. in the Conclusion, where limitation of the research is mentioned. In such respect, in fact, the adoption of a cloud-based services can improve the adoption and the usability of the framework even to not experienced people: Costantini A. et al. (2021) A Cloud-Edge Orchestration Platform for the Innovative Industrial Scenarios of the IoTwins Project. In: Gervasi O. et al. (eds) Computational Science and Its Applications – ICCSA 2021. ICCSA 2021. Lecture Notes in Computer Science, vol 12950. Springer, Cham. https://doi.org/10.1007/978-3-030-86960-1_37
Author Response
Dear Editors and Reviewers:
Thank you so much for taking the time to review our manuscript entitled “Sustainability assessment of a low-income building: A BIM-LCSA-FAHP-based analysis” (ID: buildings-1551473). We really appreciate your comments and suggestions, all of which are valid, valuable, and very helpful to us for revising and improving our paper, as well as showing us the important guiding significance of our research. We have studied the comments carefully and have made all the necessary corrections which we hope will meet with your approval.
Please refer to the revised version of the paper in which the modifications are marked in blue. The main corrections to the article and our detailed responses are below.
Responses to Editor and Reviewer Comments:
Reviewer 1:
Q1: The manuscript describe the definition of a novel framework which adopts a set of computing models to tackle different problems coming from construction industry. The study performed by the authors is relevant and the effort made is appreciated. The manuscript shows how they tackle the problem of producing new buildings, but without further burdening the environment. To do that, they adopt a defined set of computing models through a new framework which has been applyed and tested on nine scenarios, addressing different configurations of construction models. The aim of the present study is to provide a comprehensive and systemic decision-making tool that enables the best environmental, economic, and social building materials choice when planning a new construction project. The method defined, the models adopted, together with their application into the different use case and the related impact analysis are well explained.
Reply: Thank you very much for this comment. This paragraph perfectly defined the scope of work.
Q2: (i) A clear description of the computatinal resources used to test the model is missing. Today, the computing infrastructure (or computing resources and related services) for such kind of research is becaming more and more important and there is no mention in the manuscript about the resources and the services used to reach the final goal. As an example, a growing number of such activities is making use of cloud services (made available within funded projects as well as private companies).
Reply: Thanks for this suggestion. Indeed, the computational resources used were not clearly described in the initial version of the article. Please see the 1st paragraph of section 3.3 of the revised manuscript with changes marked in blue.
Section 3.3 – 1st paragraph:
Based on the pre-selected construction plan, the 3D BIM model was developed in Autodesk Revit, where technical features were entered in order to parameterize each scenario with the characteristics of the corresponding materials. Considering the typology of the analyzed building, the volume of data processed during the modeling and simulation of the case study was reduced. Therefore, the computational resource used was a Core i5 notebook, with 4GB memory, and 1TB HD. However, in future studies focusing on more complex buildings, the possibility of using a cloud service should be considered, which allows access to more robust computing infrastructure. The sustainability analysis, in turn, was developed based on different methodologies (Figure 3). For the environmental impact, Tally software was employed, an environmental impact tool that applies the Autodesk Revit API to use BIM elements with a custom LCA database (GaBi). This software combines material attributes and engineering specifications with environmental impact data (GaBi) to produce LCA reports based on the information generated by the Revit model. For the calculation of the economic impact, the Brazilian Cost and Index Research System was used. This methodology is widely employed in Brazil to determine construction costs. Finally, the social impact was calculated based on the estimated amount invested in labor for the construction of a housing unit.
Q3: The last sentence introduce to another point missing in the manuscript which is (ii) the collaboration and interaction with other group(s) or project(s), specially with those that adopt the same approach and/or develop computing-based services. To this end, to complete the information provided by the manuscript, a reference to some collaborations and/or EU funded projects running to datecan be added in terms of futhure or possible collaborations. Among the projects to consider, the IOTWINS project (https://cordis.europa.eu/project/id/857191/reporting) is aimed to provide cloud-based tools and services for (but not limeted to) the hybrid digital twins. In particular, they design a cloud-based platform services that can be exported adn implemented to other computing models. More information is described in the following publication that should becited - e.g. in the Conclusion, where limitation of the research is mentioned. In such respect, in fact, the adoption of a cloud-based services can improvethe adoption and the usability of the framework even to not experienced people: Costantini A. et al. (2021) A Cloud-Edge Orchestration Platform forthe Innovative Industrial Scenarios of the IoTwins Project. In: Gervasi O. etal. (eds) Computational Science and Its Applications – ICCSA 2021. ICCSA2021. Lecture Notes in Computer Science, vol 12950. Springer, Cham.https://doi.org/10.1007/978-3-030-86960-1_37
Reply: Thank you for pointing this out. As computational resources and simulated models become more complex, collaboration becomes essential. Reference to the IoTwins project was made in the conclusions section, as suggested. Please see the 2nd paragraph of the Conclusions section of the revised manuscript with changes marked in blue.
Conclusion – 2nd paragraph:
Although the application of this framework can contribute to increased efficiency in the execution of low-income works, this research is subjected to some limitations that should be considered, and some may serve as a stimulus for future work. First, this study does not cover other construction systems such as sewage, electrical and hydraulic systems. Measuring the impacts obtained at all stages of construction and comparing the impacts generated with the parameters obtained in this work constitute the next steps of this research. Second, the environmental analysis of reuse and recycling (stage D) was not included in this study, since the international standard presents different characteristics in the process of choosing the building materials. So, this research used the Gabi database pattern, which is included in the Tally plugin. Third, there are limitations regarding the social indicator used, as it is only related to the direct impact on the local community where the building is being constructed. Therefore, the extension of social indicators in the proposed framework is required in future work. In addition, there is no national database that allows for more reliable decision-making based on the characteristics of local materials. Thus, the creation of national and international databases is urgent and necessary, allowing a more consciously materials decision-making process. Fourth, this case study focused on the analysis of a social housing, where the volume of modeled information is limited. However, conducting research in more complex buildings may require access to advanced computing infrastructure. In this sense, collaborations with research projects that develop cloud-based platform services, such as the IoTwins project, can improve the adoption and usability of the proposed framework [89]. Finally, current research can be extended in several directions to bring the methodology closer to the real-life materials-choice process. The use of new materials options such as walls, cladding, flooring, window frames, among others, should be incorporated in future work.

Reviewer 2 Report
This paper presents the authors’ work on sustainability assessment for low-income buildings. In this study, The Triple Bottom Line (TBL) concept for sustainability was adopted, which includes three types of impacts, including environmental impact, social impact, and economic impact. Based on TBL, 9 scenarios were identified considering materials for structure, paint, and roof, and FAHP (Fuzzy Analytic Hierarchy Process) was adopted to analyze these scenarios, concluding that precast concrete, acrylic paint, and fiber cement tiles are the best option for low-income construction.
Overall, this paper is understandable with a good general structure (introduction-literature review-methodology-result-discussion-conclusion), despite some problems in language usage, also. However, after reading this manuscript, some questions related to this study have not been well addressed.
1 Low-income building is not sufficiently introduced. In the introduction section, it is not well explained why low-income buildings are investigated.
2 This study is about sustainability, but it mainly focuses on material selection, i.e., selecting materials for structure, paint, and roof. So, an introduction/literature review on the current practice for material selection is required.
3 A research gap is missing. The current literature review section is mainly about FAHP and its application. Such a literature review does not justify the proposed method (why do we need such a method). So, based on Comment 2, the authors should identify the problems (research gap) in the current practice for material selection. Otherwise, the contribution of this study cannot be assessed.
4 The method is not validated. When a new method is proposed, the authors should demonstrate that their method can work as expected and is ‘different’ from others. In this study, the authors concluded that scenario 3 (precast concrete + acrylic paint + fiber cement tiles) is the best option. However, this is not validated, e.g., via interviews with experts or other methods. This is the difference between a research paper and a project report. For a project report, you use an existing method to solve a problem and get a result, it is fine not to validate the result, but in a research paper, the result should be validated.
5 This manuscript has another feature of project reports, i.e., there is little discussion. Even though there is a section titled ‘Results and Discussion’ in the manuscript, this section is mainly about the result and its interpretation, there is little discussion.
6 Other minor problems. (a) The full version of acronyms (such as FAHP) should be provided at their first occurrence in the abstract and the main text.
Overall, this paper lacks a comprehensive literature review, which should have introduced the current practice for material selection, as well as its problems, especially for low-income buildings. And the result is not properly validated. As a result, this manuscript is more like a project report, rather than a research paper. These problems should be properly addressed.
Author Response
Reviewer 2:
Q1: This paper presents the authors’ work on sustainability assessment for low-income buildings. In this study, The Triple Bottom Line (TBL) concept for sustainability was adopted, which includes three types of impacts, including environmental impact, social impact, and economic impact. Based on TBL, 9 scenarios were identified considering materials for structure, paint, and roof, and FAHP (Fuzzy Analytic Hierarchy Process) was adopted to analyze these scenarios, concluding that precast concrete, acrylic paint, and fibercement tiles are the best option for low-income construction.
Overall, this paper is understandable with a good general structure (introduction-literature review-methodology-result-discussion-conclusion), despite some problems in language usage, also. However, after reading this manuscript, some questions related to this study have not been well addressed.
Reply: Thank you very much for this comment. The manuscript has been re-checked and we believe all errors have been corrected.
Q2: Low-income building is not sufficiently introduced. In the introduction section, it is not well explained why low-income buildings are investigated.
Reply: Thank you for pointing this out. We fully agree with this comment. A brief description of the low-income building concept has been included in the Introduction section. We also take the opportunity to include the relevance of this market in the Brazilian context, and the difficulties currently encountered during the process of choosing the most beneficial construction materials for the user and the environment. Please see the 1st and 2nd paragraphs of the Introduction section of the revised manuscript with changes marked in blue.
Introduction – 1st and 2nd paragraphs:
The construction industry is one of the most significant consumers of environmental resources worldwide [1,2], and one of the biggest responsible for giving rise to large amounts of waste [3]. The construction sector uses 30-40% of all-natural resources and primary energy over its lifespan, accounting for 30% of global greenhouse gas (GHG) emissions and representing about 6% of the world’s Gross Domestic Product (GDP) [4,5]. Despite the sector’s relevance in the international economic scenario, the United Nations (UN) estimates that about one billion people globally still live in inadequate buildings or do not have a place to live [6]. Access to adequate housing is an internationally recognized basic human right that plays an important role in society, providing citizens with dignity and security [7,8]. In Brazil, the housing deficit is approximately 6.2 million dwellings, which mainly affects low-income families [9-11]. Therefore, in an attempt to mitigate this social problem and provide access to housing for the portion of the population traditionally excluded from the conventional real estate market, the Brazilian Federal Government created the program designated “Minha Casa, Minha Vida” (My House, My Life), aimed at simplifying, financing, and encouraging the construction and acquisition of low-cost houses [10,12,13].
Low-income buildings or social housing can be defined as dwellings built on a large scale with government funding aimed at the low-income population [9]. However, due to the growing demand for this type of building, decision-makers are often pressured to speed up the construction process and reduce costs [10]. As a result, there is a tendency to use lower-cost materials, which are not always the most beneficial choices for the user and the environment [11]. Thus, it is essential to develop material selection tools that help decision-making in a transparent, reliable, and sustainable way. In this context, faced with the need to produce new buildings, but without further burdening the environment, attempts to improve social, economic, and environmental indicators have turned attention to building construction. These efforts have focused on complying with the pillars of the theory known as Triple Bottom Line (TBL), which considers that development only actually occurs when the best use of natural resources, the guarantee or improvement of the current economic balance, and the occurrence of social gains are observed concomitantly and with parity [14,15]. Unfortunately, this strategy is not always used in the construction market. Environmental and social aspects are usually neglected, and relevant decisions are taken considering mainly the economic aspects, which are more easily measured.
Q3: This study is about sustainability, but it mainly focuses on material selection, i.e., selecting materials for structure, paint, and roof. So, an introduction/literature review on the current practice for material selection is required.
Reply: Thanks to the Reviewer for this suggestion in improving the manuscript. We agree that more emphasis could have been placed on the current practice for material selection, so we have added the following information in paragraphs 1 and 2 of the Literature Review section as a way to mitigate this shortcoming.
Literature Review – 1st and 2nd paragraphs:
The impact caused by the construction industry on the environment, economy, and society is significant [23]. The large number of resources required for the construction and retrofit of buildings gradually brought sustainable construction into focus [24]. In this sense, the selection of suitable and sustainable building materials during the design stage has a direct impact on the entire life cycle of the building [25,26], promoting the reduction of its cost, allowing a better use of natural resources, reducing the generation of waste during construction and operation stages, optimizing energy consumption, and providing a healthier environment for users [27,28]. Thereby, several researchers conclude that the correct selection of materials is one of the simplest and most efficient ways to incorporate sustainability in the construction industry [25,29,30]. Despite the relevance of optimizing the construction materials selection process, currently, in practice, this is still primarily done empirically, based on the engineers' knowledge and experience, rather than numerical approaches [30-32]. However, when associated with the lack of experience of the responsible professional, this heuristic approach prevents the available alternatives from being objectively compared [25,33], and the lowest price ends up prevailing as the most important choice criterion. Therefore, it is essential to develop a systematic decision model that helps team members to choose the most advantageous materials, based on well-defined, sustainable, and unambiguous criteria [23,32].
In order to overcome this problem, several methodological approaches have been employed to deal with the material selection problem [24,27]. Some recent studies have been using the Building Research Institute Environmental Assessment Method (BREEAM) as a construction material selection tool, while others propose the use of the Leadership in Energy and Environmental Design (LEED) rating system [34,35]. However, Life Cycle Assessment (LCA) is still the most applied methodology to assess the impacts of a product throughout its entire life cycle [36]. Takano et al. [37] used LCA to research the influence of material selection on the energy balance of a building. Najjar et al. [20] studied the performance of building materials using a BIM-LCA approach. Gardner et al. [38] conducted a material LCA in a building comparing the impacts of each stage of the construction process. Röck et al. [39] used a BIM-LCA-integrated approach to compare building elements contribution to the total embodied environmental impact. Abd Rashid et al., [40] evaluated a cradle-to-grave environmental impact of a residential building in Malaysia through LCA. However, LCA-based models focus on evaluating the environmental performance of a product, not adequately considering issues related to its costs and society [24,30,41], making it necessary to develop new methods that overcome these flaws, integrating the three pillars of sustainable development.
Q4: A research gap is missing. The current literature review section is mainly about FAHP and its application. Such a literature review does not justify the proposed method (why do we need such a method). So, based on Comment 2, the authors should identify the problems (research gap) in the current practice for material selection. Otherwise, the contribution of this study cannot be assessed.
Reply: We thank the Reviewer for this suggestion. We agree with this and, in addition to having paid more attention to the current practice of material selection (as per the previous comment – Q3), we rewrote the 7th paragraph of the Literature Review section to highlight the research gap. We also restructured the 6th paragraph of the Literature Review section to include only studies that used the AHP or FAHP for material selection. With that, we believe that the literature review justifies the proposed method, and we hope it meets with your approval.
Literature Review – 7th paragraph:
Although there is consensus on the importance of establishing criteria and processes for the selection of suitable construction materials, few studies have been carried out to assess the sustainability of the materials considering the TBL methodology [76-78]. An even smaller number employ MCDM techniques associated with computer BIM software such as Revit and Tally [5]. Tally is a Revit plugin used to perform LCA of building materials, where materials parameterized in the BIM model are transported to an existing Life Cycle Inventory (LCI) database, collecting input and output data, including material consumption raw materials, water, energy, and the emission of solid, gaseous and liquid waste [5,79]. To fill the gap in existing research, this study proposes a BIM-LCSA-FAHP-based analysis, in order to develop a framework to select optimal building materials. In addition to the tools mentioned, this study will use the GaBi database, which is considered one of the most complete tools for organizing inventory data [80] and perform impact assessments [81].
Literature Review – 6th paragraph:
Relevant studies have been developed using FAHP in materials selection problems. Figueiredo et al. [44] proposed a decision-making framework using FAHP to facilitate the choice of building materials. Akadiri et al. [30] selected sustainable building materials using the Fuzzy Extended Analytic Hierarchy Process (FEAHP). Tian et al. [73] combined AHP and gray-correlation TOPSIS to select sustainable decor materials. Singh et al. [74] integrated FAHP and TOPSIS in the selection of composite materials applied to structural elements. A similar strategy was presented by Janowska-Renkas et al. [32], who used FEAHP and Fuzzy-TOPSIS to select the best high-performance concrete for post-tension bridges. Mayhoub et al. [27] applied AHP in the development of a new evaluation framework based on four green building rating systems for selecting sustainable façade materials. A similar study was conducted by Ruslan et al. [33], who used AHP and Value-Based Analysis to choose the most sustainable material for façade use, between Aluminum Composite Panel (ACP) and Stainless Steel. Dinh et al. [25] identified a list of 18 sustainability criteria applicable to the selection of sustainable materials in Vietnam, and used AHP to rank them based on their degree of relevance to the country's construction sector. Lee et al. [31] introduced a new AHP-based approach to selecting building materials based on their performance, and applied it to a case study of a concrete formwork system. Finally, Ogrodnik [75] performed a comprehensive review of MCDM methods and concluded that AHP is one of the most popular techniques currently used.
Below is the insertion of the new papers in the references section:
- Mercader-Moyano, P.; Esquivias, P.M.,; Muntean, R. Eco-Efficient analysis of a refurbishment proposal for a social housing. Sustainability 2020, 12, 6725, doi: https://doi.org/10.3390/su12176725
- Carnemolla, P.; Skinner, V. Outcomes associated with providing secure, stable, and permanent housing for people who have been homeless: An international scoping review. Plann. Lit. 2021, 36(4), 508–525, doi: https://doi.org/10.1177/05854122211012911
- Oliveira, R.; Vicente, R.; Almeida, R.M.S.F.; Figueiredo, A. The importance of in situ characterisation for the mitigation of poor indoor environmental conditions in social housing. Sustainability 2021, 13, 9836, doi: https://doi.org/10.3390/su13179836
- Santos, L.; Pontes, I.S.; Bastos, L.P.; Melo, G.V.M.; Barata, M. Acoustic perfomance of social housings in Brazil: Assessment of light weight expanded polystyrene concrete as resilient subfloor. Build. Eng. 2021, 41, 102442, doi: https://doi.org/10.1016/j.jobe.2021.102442
- Dalbem, R.; Cunha, E.G.; Vicnete, R.; Figueiredo, A.; Oliveira, R.; Silva, A.C.S.B. Optimisation of a social housing for south of Brazil: From basic performance standard to passive house concept. 2019, 167, 1278-1296, doi: https://doi.org/10.1016/j.energy.2018.11.053
- Mahecha, R.E.G.; Caldas, L.R.; Garaffa, R.; Lucena, A.F.P.; Szklo, A.; Filho, R.D.T. Constructive systems for social housing deployment in developing countries: A case study using dynamic life cycle carbon assessment and cost analusis in Brazil. Energy Build. 2020, 227, 110395, doi: https://doi.org/10.1016/j.enbuild.2020.110395
- Paidakaki, A.; Lang, R. Uncovering social sustainability in housing systems through the lens of institutional capital: A study of two housing alliances in Vienna, Austria. Sustainability 2021, 13, 9726, doi: https://doi.org/10.3390/su13179726
- Rossi, M.M.; Favretto, A.P.O.; Grassi, C.; DeCarolis, J.; Cho, S.; Hill, D.; Chvatal, K.M.S.; Ranjithan, R. Metamodels to assess the thermal perfomance of naturally ventilated, low-cost houses in Brazil. Energy Build. 2019, 204, 109457, doi: https://doi.org/10.1016/j.enbuild.2019.109457
- Al-atesh, E.A.; Rahmawati, Y.; Zawawi, N.A.W.A.; Elmansoury, A. Developing the green building materials selection criteria for sustainable building projects. J. Adv. Sci. Eng. Inf. Technol. 2021, 5, 2112-2120.
- Chen, Z.; Yang, L.; Chin, K.; Pedrycz, W.; Chang, J.; Martínez, L.; Skibniewski, M.J. Sustainable building material selection: An integrated multi-criteria large group decision making framework. Soft Comput. 2021, 113, 107903, doi: https://doi.org/10.1016/j.asoc.2021.107903
- Dinh, T.H.; Dinh, T.H.; Götze, U. Integration of sustainability criteria and life cycle sustainability assessment method into construction material selection in developing countries: The case of Vietnam. J. Sustainable Dev. Plann. 2020, 15(8), 1145-1156, doi: https://doi.org/10.18280/ijsdp.150801
- Kanniyapan, G.; Nesan, L.J.; Mohammad, I.S.; Keat, T.S.; Ponniah, V. Selection criteria of building material for optimising maintainability. Build. Mater. 2019, 221, 651-660, doi: https://doi.org/10.1016/j.conbuildmat.2019.06.108
- Mayhoub, M.M.G.; El Sayad, Z.M.T.; Ali. A.A.M.; Ibrahim, M.G. Assessment of green building materials’ attributes to achieve sustainable building façades using AHP. Buildings 2021, 11, 474, doi: https://doi.org/10.3390/buildings11100474
- Hatefi, S.M.; Asadi, H.; Shams, G.; Tamosaitien, J.; Turkis, Z. Model for the sustainable material selection by applying integrated dempster-shafer evidence theory and additive ratio assessment (ARAS) method. Sustainability 2021, 13, 10438, doi: https://doi.org/10.3390/su131810438
- Chen, Z.; Martínez, L.; Chang, J.; Wang, X.; Xionge, S.; Chin, K. Sustainable building material selection: A QFD- and ELECTRE III-embedded hybrid MCGDM approach with consensus building. Appl. Artif. Intell. 2019, 85, 783-807, doi: https://doi.org/10.1016/j.engappai.2019.08.006
- Akadiri, P.O.; Olomolaiye, P.O.; Chinyio, E.A. Multi-criteria evaluation model for the selection of sustainable materials for building projects. Constr. 2013, 30, 113–125, doi: https://doi.org/10.1016/j.autcon.2012.10.004
- Lee, D.; Lee, D.; Lee, M.; Kim, M.; Kim, T. Analytic hierarchy process-based construction material selection for performance improvement of building construction: The case of a concrete system form. Materials 2020, 13, 1738, doi: https://doi.org/10.3390/ma13071738
- Janowska-Renkas, E.; Jakiel, P.; Fabianowski, D.; Matyjaszczyk, D. Optimal selection of high-performance concrete for post-tensioned girder bridge using advanced hybrid MCDA method. Materials 2021, 14, 6553, doi: https://doi.org/10.3390/ma14216553
- Ruslan, A.A.B.; Al-atesh, E.A.; Rahmawati, Y.; Utomo, C.; Zawawi, N.A.W.A.; Jahja, M.; Elmansoury, A. A value-based decision-making model for selecting sustainable materials for buildings. J. Adv. Sci. Eng. Inf. Technol. 2021, 6, 2279-2286.
- Castro-Lacouture, D.; Sefair, J.A.; Medaglia, A.L. Optimization model for the selection of materials using LEED-based green building rating system in Colombia. Environ. 2009, 44(6), 1162-1170, doi: https://doi.org/10.1016/j.buildenv.2008.08.009
- Marzouk, M.; Azab, S.; Metawie, M. BIM-based approach for optimizing life cycle costs of sustainable buildings. Cleaner Prod. 2018, 188, 217-226, doi: https://doi.org/10.1016/j.jclepro.2018.03.280
- Nawarathna, A.; Siriwardana, M.; Alwan, Z. Embodied carbon as a material selection criterion: insights from Sri Lankan construction sector. Sustainability 2021, 13,2202, doi: https://doi.org/10.3390/su13042202
- Takano, A.; Pal, S.K.; Kuittinen, M.; Alanne, K.; Hughes, M.; Winter, S. The effect of material selection on life cycle energy balance: A case study on a hypothetical building model in Filand. Environ. 2015, 89, 192-202, doi: https://doi.org/10.1016/j.buildenv.2015.03.001
- Gardner, H.; Garcia, J.; Hasik, V.; Olinzock, M.; Banawi, A.; Bilec, M.M. Materials life cycle assessment of a living building. Procedia CIRP 2019, 80, 458-463, doi: https://doi.org/10.1016/j.procir.2019.01.021
- Röck, M.; Hollberg, A.; Habert, G.; Passer, A. LCA and BIM: Visualization of environmental potentials in building construction at early design stages. Environ. 2018, 140, 153-161, doi: https://doi.org/10.1016/j.buildenv.2018.05.006
- Abd Rashid, A.F.; Idris, J.; Yusoff, S. Environmental impact analysis on residential building in Malaysia using life cycle assessment. Sustainability 2017, 9(3), 329, doi: https://doi.org/10.3390/su9030329
- Cabeza, L.F.; Rincón, L.; Vilariño, V.; Pérez, G.; Castell, A. Life cycle assessment (LCA) and life cycle energy analysis (LCEA) of buildings and the building sector: A review. Renewable Sustainable Energy Rev. 2014, 29, 394-416, doi: https://doi.org/10.1016/j.rser.2013.08.037
- Kloepffer, W. Life cycle sustainability assessment of products. J. Life Cycle Assess. 2008, 13, 89, doi: https://doi.org/10.1065/lca2008.02.376
- Yang, W.; Chon, S.; Choe, C.; Yang, J. Materials selection method using TOPSIS with some popular normalization methods. Res. Express 2021, 3, 015020, doi: https://doi.org/10.1088/2631-8695/abd5a7
- Tian, G.; Zhang, H.; Feng, Y.; Wang, D.; Peng, Y.; Jia, H. Green decoration materials selection under interior environment characteristics: A grey-correlational based hybrid MCDM method. Renewable Sustainable Energy Rev. 2018, 81(1), 682-692, doi: https://doi.org/10.1016/j.rser.2017.08.050
- Singh, A.K.; Avikal, s.; Kumar, N.; Kumar, M.; Thakura, P. A fuzzy-AHP and M-TOPSIS based approach for selection of composite materials used in structural applications. Today:. Proc. 2020, 26(2), 3119-3123, doi: https://doi.org/10.1016/j.matpr.2020.02.644
- Mathiyazhagan, K.; Gnanavelbabu, A.; Prabhuraj, B.L. A sustainable assessment model for material selection in construction industries perspective using hybrid MCDM approaches. Adv. Manage. Res. 2019, 16(2), 234-259, doi: https://doi.org/10.1108/JAMR-09-2018-0085
- Zhou, C.; Yin, G.; Hu, X. Multi-objective optimization of material selection for sustainable products: Artificial neural networks and genetic algorithm approach. Des. 2009, 30(4), 1209-1215, doi: https://doi.org/10.1016/j.matdes.2008.06.006
- Akadiri, P.O. Understanding barriers affecting the selection of sustainable materials in building projects. Build. Eng. 2015, 4, 86-93, doi: https://doi.org/10.1016/j.jobe.2015.08.006
- Constantini, A.; Duma, D.C.; Martelli, B.; Antonacci, M.; Galletti, M.; Tisbenim S.R.; Bellavista, P.; Di Modica, G.; Nehls, D.; Ahouangonou, J.; Delamarre, C.; Cesini, D. A cloud-edge orchestration platform for the innocative industrial scenarios of the IoTwins project. In: Gervasi O. et al. (eds) Computational Science and Its Applications – ICCSA 2021. Lecture Notes in Computer Science 2021, 12950, Springer. doi: https://doi.org/10.1007/978-3-030-86960-1_37
Q5: The method is not validated. When a new method is proposed, the authors should demonstrate that their method can work as expected and is ‘different’ from others. In this study, the authors concluded that scenario 3 (precast concrete + acrylic paint + fiber cement tiles) is the best option. However, this is not validated, e.g., via interviews with experts or other methods. This is the difference between a research paper and a project report. For a project report, you use an existing method to solve a problem and get a result, it is fine not to validate the result, but in a research paper, the result should be validated.
Reply: Thank you very much for this comment. We believe that the best validation for this case study would be through an assessment of a real building, which this group intends to carry out in a future study. However, for this, we understand that the methodological approach needs to be previously well defined. Therefore, the objective of this article is the development of this methodology through the combination of techniques already validated by previous publications, but separately. Although the case study was chosen because it is representative of the reality of Brazilian low-income buildings, the method can be applied in different contexts and cultures, through the variation of the weights of the criteria, the costs of materials, and the environmental impacts related to each product. In this context, there was a great concern to present a methodological path that could be replicated for any type of construction, considering the reality of any country. Please see the 3rd paragraph of the Materials and Methods section, and paragraphs 2, 3, and 4, of the Discussion section in the revised manuscript, where we try to improve this matter.
Materials and Methods:
In order to validate this proposal, a case study will be presented in the following sections. This validation through a case study intends to present the applicability of the method and discuss how this methodological framework can be helpful to benefit the project decision-making process. In this work, the case study was chosen to be representative of the reality of Brazilian low-income construction, although the method can be applied in different contexts and cultures.
Discussion – 2nd, 3rd and 4th paragraphs:
It is essential to point out that the methodological framework proposed in this study was tested for a proposed building. This is valid since many publications in the literature present similar validations, but it brings certain limitations to the study compared to an actual building. For example, regarding the economic criterion analyzed, information from the Brazilian Cost and Index Research System was considered instead of collecting data in the field (i.e., price surveys in stores in the construction region). The same happens for the social criterion since the real impact of a building of this size on society was not investigated in the region. In an actual project, the weights of the criteria could be decided, for example, through questionnaires made available to all those directly or indirectly involved in the building construction and operation.
Besides, when applied to real case studies, impact categories should be chosen according to stakeholder needs and priorities. Environmental impact categories, for example, can be chosen concerning what is most critical in the region. However, this work aims to show the importance and applicability of considering the three pillars of sustainability during any building assessment. Even though the weights of the criteria can vary drastically from one project to another, the TBL approach must be applied to ensure the development of more sustainable construction industry.
There was a great concern to present a methodological path that could be replicated for any type of construction, considering the reality of any country. Therefore, the authors understand that the objective of the analysis has been achieved by presenting a proposal that different professionals can apply to improve the decision-making process of which building materials are more sustainable in each case. Finally, it is worth mentioning that the case study presented focuses on the construction of low-income buildings, which is a subject that still lacks much discussion in the literature. With the proposed methodological framework, it is clear that it is possible to consider the three pillars of sustainability together, even when the project budget is short.
Q6: This manuscript has another feature of project reports, i.e., there is little discussion. Even though there is a section titled ‘Results and Discussion’ in the manuscript, this section is mainly about the result and its interpretation, there is little discussion.
Reply: Thank you very much for this comment. We restructured Section 4, which is now focused on results only, and created Section 5, with four discussion paragraphs.
Discussion:
The application of the proposed method, integrating LCSA, BIM and MCDM, demonstrated that scenario 3 was considered the most sustainable option for the analyzed building. It is interesting to highlight that among the three categories of materials analyzed, those that presented the best results for this case study were: precast concrete (structural material), acrylic paint (painting), and fiber-cement tiles (roofing). Identifying and modeling the implications of individual choices has always been fundamental to building construction. In this context, scenario 3 was ranked the third lowest cost, the lowest environmental impact, and the third-highest positive social impact, making this scenario the best option for this type of low-income housing unit.
It is essential to point out that the methodological framework proposed in this study was tested for a proposed building. This is valid since many publications in the literature present similar validations, but it brings certain limitations to the study compared to an actual building. For example, regarding the economic criterion analyzed, information from the Brazilian Cost and Index Research System was considered instead of collecting data in the field (i.e., price surveys in stores in the construction region). The same happens for the social criterion since the real impact of a building of this size on society was not investigated in the region. In an actual project, the weights of the criteria could be decided, for example, through questionnaires made available to all those directly or indirectly involved in the building construction and operation.
Besides, when applied to real case studies, impact categories should be chosen according to stakeholder needs and priorities. Environmental impact categories, for example, can be chosen concerning what is most critical in the region. However, this work aims to show the importance and applicability of considering the three pillars of sustainability during any building assessment. Even though the weights of the criteria can vary drastically from one project to another, the TBL approach must be applied to ensure the development of more sustainable construction industry.
There was a great concern to present a methodological path that could be replicated for any type of construction, considering the reality of any country. Therefore, the authors understand that the objective of the analysis has been achieved by presenting a proposal that different professionals can apply to improve the decision-making process of which building materials are more sustainable in each case. Finally, it is worth mentioning that the case study presented focuses on the construction of low-income buildings, which is a subject that still lacks much discussion in the literature. With the proposed methodological framework, it is clear that it is possible to consider the three pillars of sustainability together, even when the project budget is short.
Q7: Other minor problems. (a) The full version of acronyms (such as FAHP) should be provided at their first occurrence in the abstract and the main text.
Reply: Thanks for this comment. In the abstract, the first occurrence of the terms "BIM", "LCSA" and "FAHP" is in the model description ("BIM-LCSA-FAHP-based model"). Therefore, to facilitate reading, we chose to keep the acronyms at this first moment. However, all complete versions of these three acronyms are provided in the following lines of the abstract itself. We also re-checked the manuscript and we believe that all full versions of acronyms are now provided on the first occurrence in the main text.

Round 2
Reviewer 2 Report
Thanks for providing the revised manuscript. All my previous concerns have been properly addressed. The rationales for investigating the low-income buildings have been sufficiently explained in the introduction, a comprehensive literature review has been conducted to justify the proposed methodology and the contribution of the study, and the discussion section has been expanded.
Just one more minor comment:
To briefly explain the relationship between ‘sustainability’ and the ‘Triple Bottom Line’ using one or two sentences.
Author Response
Reviewer 2:
Q1: Thanks for providing the revised manuscript. All my previous concerns have been properly addressed. The rationales for investigating the low-income buildings have been sufficiently explained in the introduction, a comprehensive literature review has been conducted to justify the proposed methodology and the contribution of the study, and the discussion section has been expanded.
Just one more minor comment:
To briefly explain the relationship between ‘sustainability’ and the ‘Triple Bottom Line’ using one or two sentences.
Reply: Thank you for the time and effort you put into improving our paper. A brief description of sustainability and the Triple Bottom Line concepts has been included in the Introduction section. Please see the 2nd paragraph of Introduction section of the revised manuscript with changes marked in blue.
Introduction – 2nd paragraph:
Low-income buildings or social housing can be defined as dwellings built on a large scale with government funding aimed at the low-income population [9]. However, due to the growing demand for this type of building, decision-makers are often pressured to speed up the construction process and reduce costs [10]. As a result, there is a tendency to use lower-cost materials, which are not always the most beneficial choices for the user and the environment [11]. Thus, it is essential to develop material selection tools that help decision-making in a transparent, reliable, and sustainable way. In this context, faced with the need to produce new buildings, but without further burdening the environment, attempts to improve social, economic, and environmental indicators have turned attention to building construction. These efforts have focused on complying with the pillars of the theory known as Triple Bottom Line (TBL), which considers that development only actually occurs when the best use of natural resources, the guarantee or improvement of the current economic balance, and the occurrence of social gains are observed concomitantly and with parity [14,15]. This theory, formulated more than two decades ago, gained notoriety by providing a framework for evaluating the sustainability of products and services based on criteria other than just the environmental one. Currently, additional dimensions have been added to the original concept, such as the cultural pillar and the political pillar, but the original tripod still symbolizes the basic elements to be considered in a sustainability assessment. Unfortunately, this strategy is not always used in the construction market. Environmental and social aspects are usually neglected, and relevant decisions are taken considering mainly the economic aspects, which are more easily measured.
